# Controllable Text-to-Image Generation

**Bowen Li, Xiaojuan Qi, Thomas Lukasiewicz, Philip H. S. Torr**
University of Oxford
{bowen.li, thomas.lukasiewicz}@cs.ox.ac.uk
{xiaojuan.qi, philip.torr}@eng.ox.ac.uk

## Abstract

In this paper, we propose a novel controllable text-to-image generative adversarial network (ControlGAN), which can effectively synthesise high-quality images and also control parts of the image generation according to natural language descriptions. To achieve this, we introduce a word-level spatial and channel-wise attention-driven generator that can disentangle different visual attributes, and allow the model to focus on generating and manipulating subregions corresponding to the most relevant words. Also, a word-level discriminator is proposed to provide fine-grained supervisory feedback by correlating words with image regions, facilitating training an effective generator which is able to manipulate specific visual attributes without affecting the generation of other content. Furthermore, perceptual loss is adopted to reduce the randomness involved in the image generation, and to encourage the generator to manipulate specific attributes required in the modified text. Extensive experiments on benchmark datasets demonstrate that our method outperforms existing state of the art, and is able to effectively manipulate synthetic images using natural language descriptions. Code is available at https://github.com/mrlibw/ControlGAN.

## 1   Introduction

Generating realistic images that semantically match given text descriptions is a challenging problem and has tremendous potential applications, such as image editing, video games, and computer-aided design. Recently, thanks to the success of generative adversarial networks (GANs) [4, 6, 15] in generating realistic images, text-to-image generation has made remarkable progress [16, 25, 27] by implementing conditional GANs (cGANs) [5, 16, 17], which are able to generate realistic images conditioned on given text descriptions.

However, current generative networks are typically uncontrollable, which means that if users change some words of a sentence, the synthetic image would be significantly different from the one generated from the original text as shown in Fig. 1. When the given text description (e.g., colour) is changed, corresponding visual attributes of the bird are modified, but other unrelated attributes (e.g., the pose and position) are changed as well. This is typically undesirable in real-world applications, when a user wants to further modify the synthetic image to satisfy her preferences.

The goal of this paper is to generate images from text, and also allow the user to manipulate synthetic images using natural language descriptions, in one framework. In particular, we focus on modifying visual attributes (e.g., category, texture, and colour) of objects in the generated images by changing given text descriptions. To achieve this, we propose a novel controllable text-to-image generative adversarial network (ControlGAN), which can synthesise high-quality images, and also allow the user to manipulate objects' attributes, without affecting the generation of other content.

Our ControlGAN contains three novel components. The first component is the word-level spatial and channel-wise attention-driven generator, where an attention mechanism is exploited to allow the

This bird has a **yellow** back and rump, **gray** outer rectrices, and a light **gray** breast. (original text)

This bird has a **red** back and rump, **yellow** outer rectrices, and a light **white** breast. (modified text)

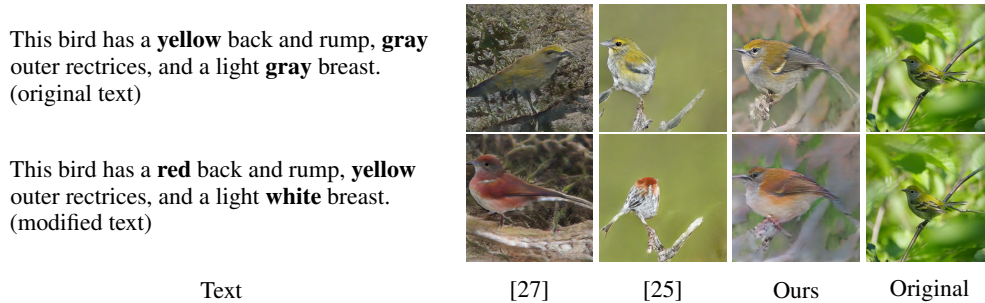

Text　　　　[27]　　　[25]　　　Ours　　Original

Figure 1: Examples of modifying synthetic images using a natural language description. The current state of the art methods generate realistic images, but fail to generate plausible images when we slightly change the text. In contrast, our method allows parts of the image to be manipulated in correspondence to the modified text description while preserving other unrelated content.

generator to synthesise subregions corresponding to the most relevant words. Our generator follows a multi-stage architecture [25, 28] that synthesises images from coarse to fine, and progressively improves the quality. The second component is a word-level discriminator, where the correlation between words and image subregions is explored to disentangle different visual attributes, which can provide the generator with fine-grained training signals related to visual attributes. The third component is the adoption of the perceptual loss [7] in text-to-image generation, which can reduce the randomness involved in the generation, and enforce the generator to preserve visual appearance related to the unmodified text.

To this end, an extensive analysis is performed, which demonstrates that our method can effectively disentangle different attributes and accurately manipulate parts of the synthetic image without losing diversity. Also, experimental results on the CUB [23] and COCO [10] datasets show that our method outperforms existing state of the art both qualitatively and quantitatively.

## 2　Related Work

**Text-to-image Generation.**　Recently, there has been a lot of work and interest in text-to-image generation. Mansimov et al. [11] proposed the AlignDRAW model that used an attention mechanism over words of a caption to draw image patches in multiple stages. Nguyen et al. [13] introduced an approximate Langevin approach to synthesise images from text. Reed et al. [16] first applied the cGAN to generate plausible images conditioned on text descriptions. Zhang et al. [27] decomposed text-to-image generation into several stages generating image from coarse to fine. However, all above approaches mainly focus on generating a new high-quality image from a given text, and cannot allow the user to manipulate the generation of specific visual attributes using natural language descriptions.

**Image-to-image translation.**　Our work is also closely related to conditional image manipulation methods. Cheng et al. [3] produced high-quality image parsing results from verbal commands. Zhu et al. [31] proposed to change the colour and shape of an object by manipulating latent vectors. Brock et al. [2] introduced a hybrid model using VAEs [9] and GANs, which achieved an accurate reconstruction without loss of image quality. Recently, Nam et al. [12] built a model for multi-modal learning on both text descriptions and input images, and proposed a text-adaptive discriminator which utilised word-level text-image matching scores as supervision. However, they adopted a global pooling layer to extract image features, which may lose important fine-grained spatial information. Moreover, the above approaches focus only on image-to-image translation instead of text-to-image generation, which is probably more challenging.

**Attention.**　The attention mechanism has shown its efficiency in various research fields including image captioning [24, 30], machine translation [1], object detection [14, 29], and visual question answering [26]. It can effectively capture task-relevant information and reduce the interference from less important one. Recently, Xu et al. [25] built the AttnGAN model that designed a word-level spatial attention to guide the generator to focus on subregions corresponding to the most relevant word. However, spatial attention only correlates words with partial regions without taking channel

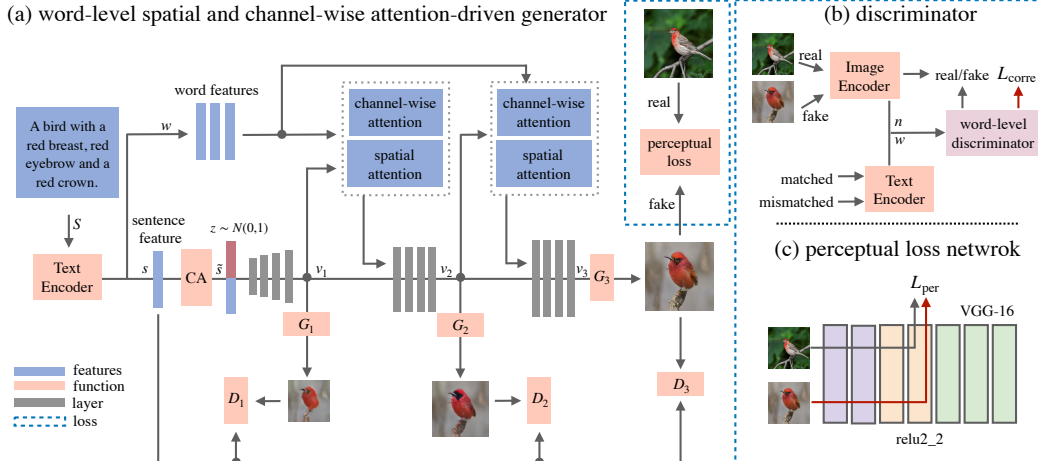

Figure 2: The architecture of our proposed ControlGAN. In (b), $\mathcal{L}_{\text{corre}}$ is the correlation loss discussed in Sec. 3.3. In (c), $\mathcal{L}_{\text{per}}$ is the perceptual loss discussed in Sec. 3.4.

information into account. Also, different channels of features in CNNs may have different purposes, and it is crucial to avoid treating all channels without distinction, such that the most relevant channels in the visual features can be fully exploited.

# 3   Controllable Generative Adversarial Networks

Given a sentence $S$, we aim to synthesise a realistic image $I'$ that semantically aligns with $S$ (see Fig. 2), and also make this generation process controllable – if $S$ is modified to be $S_m$, the synthetic result $\tilde{I}'$ should semantically match $S_m$ while preserving irrelevant content existing in $I'$ (shown in Fig. 4). To achieve this, we propose three novel components: 1) a channel-wise attention module, 2) a word-level discriminator, and 3) the adoption of the perceptual loss in the text-to-image generation. We elaborate our model as follows.

## 3.1   Architecture

We adopt the multi-stage AttnGAN [25] as our backbone architecture (see Fig. 2). Given a sentence $S$, the text encoder – a pre-trained bidirectional RNN [25] – encodes the sentence $S$ into a sentence feature $s \in \mathbb{R}^D$ with dimension $D$ describing the whole sentence, and word features $w \in \mathbb{R}^{D \times L}$ with length $L$ (i.e., number of words) and dimension $D$. Following [27], we also apply conditioning augmentation (CA) to $s$. The augmented sentence feature $\tilde{s}$ is further concatenated with a random vector $z$ to serve as the input to the first stage. The overall framework generates an image from coarse-to fine-scale in multiple stages, and, in each stage, the network produces a hidden visual feature $v_i$, which is the input to the corresponding generator $G_i$ to produce a synthetic image. Spatial attention [25] and our proposed channel-wise attention modules take $w$ and $v_i$ as inputs, and output attentive word-context features. These attentive features are further concatenated with the hidden feature $v_i$ and then serve as input for the next stage.

The generator exploits the attention mechanism via incorporating a spatial attention module [25] and the proposed channel-wise attention module. The spatial attention module [25] can only correlate words with individual spatial locations without taking channel information into account. Thus, we introduce a channel-wise attention module (see Sec. 3.2) to exploit the connection between words and channels. We experimentally find that the channel-wise attention module highly correlates semantically meaningful parts with corresponding words, while the spatial attention focuses on colour descriptions (see Fig. 6). Therefore, our proposed channel-wise attention module, together with the spatial attention, can help the generator disentangle different visual attributes, and allow it to focus only on the most relevant subregions and channels.

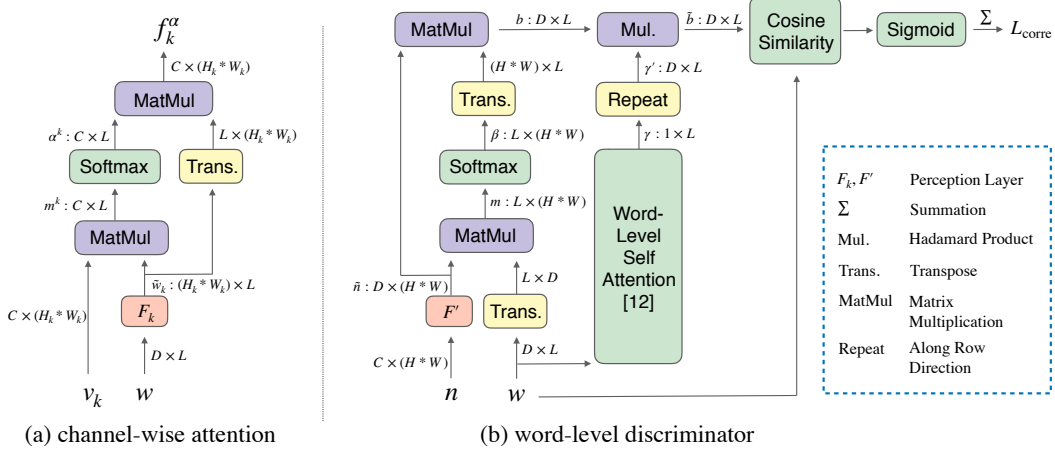

(a) channel-wise attention        (b) word-level discriminator

Figure 3: The architecture of proposed channel-wise attention module and word-level discriminator.

## 3.2 Channel-Wise Attention

At the $k^{th}$ stage, the channel-wise attention module (see Fig. 3 (a)) takes two inputs: the word features $w$ and hidden visual features $v_k \in \mathbb{R}^{C \times (H_k * W_k)}$, where $H_k$ and $W_k$ denote the height and width of the feature map at stage $k$. The word features $w$ are first mapped into the same semantic space as the visual features $v_k$ via a perception layer $F_k$, producing $\tilde{w}_k = F_k w$, where $F_k \in \mathbb{R}^{(H_k * W_k) \times D}$.

Then, we calculate the channel-wise attention matrix $m^k \in \mathbb{R}^{C \times L}$ by multiplying the converted word features $\tilde{w}_k$ and visual features $v_k$, denoted as $m^k = v_k \tilde{w}_k$. Thus, $m^k$ aggregates correlation values between channels and words across all spatial locations. Next, $m^k$ is normalised by the softmax function to generate the normalised channel-wise attention matrix $\alpha^k$ as

$$\alpha_{i,j}^k = \frac{\exp(m_{i,j}^k)}{\sum_{l=0}^{L-1} \exp(m_{i,l}^k)}. \tag{1}$$

The attention weight $\alpha_{i,j}^k$ represents the correlation between the $i^{th}$ channel in the visual features $v_k$ and the $j^{th}$ word in the sentence $S$, and higher value means larger correlation.

Equipped with the channel-wise attention matrix $\alpha^k$, we obtain the final channel-wise attention features $f_k^\alpha \in \mathbb{R}^{C \times (H_k * W_k)}$, denoted as $f_k^\alpha = \alpha^k (\tilde{w}_k)^T$. Each channel in $f_k^\alpha$ is a dynamic representation weighted by the correlation between words and corresponding channels in the visual features. Thus, channels with high correlation values are enhanced resulting in a high response to corresponding words, which can facilitate disentangling word attributes into different channels, and also reduce the influence from irrelevant channels by assigning a lower correlation.

## 3.3 Word-Level Discriminator

To encourage the generator to modify only parts of the image according to the text, the discriminator should provide the generator with fine-grained training feedback, which can guide the generation of subregions corresponding to the most relevant words. Actually, the text-adaptive discriminator [12] also explores the word-level information in the discriminator, but it adopts a global average pooling layer to output a 1D vector as image feature, and then calculates the correlation between image feature and each word. By doing this, the image feature may lose important spatial information, which provides crucial cues for disentangling different visual attributes. To address the issue, we propose a novel word-level discriminator inspired by [12] to explore the correlation between image subregions and each word; see Fig. 3 (b).

Our word-level discriminator takes two inputs: 1) word features $w, w'$ encoded from the text encoder, which follows the same architecture as the one (see Fig. 2 (a)) used in the generator, where $w$ and $w'$ denote word features encoded from the original text $S$ and a randomly sampled mismatched text, respectively, and 2) visual features $n_{\text{real}}, n_{\text{fake}}$, both encoded by a GoogleNet-based [22] image encoder from the real image $I$ and generated images $I'$, respectively.

For simplicity, in the following, we use $n \in \mathbb{R}^{C \times (H*W)}$ to represent visual features $n_{\text{real}}$ and $n_{\text{fake}}$, and use $w \in \mathbb{R}^{D \times L}$ for both original and mismatched word features. The word-level discriminator contains a perception layer $F'$ that is used to align the channel dimension of visual feature $n$ and word feature $w$, denoted as $\tilde{n} = F'n$, where $F' \in \mathbb{R}^{D \times C}$ is a weight matrix to learn. Then, the word-context correlation matrix $m \in \mathbb{R}^{L \times (H*W)}$ can be derived via $m = w^T \tilde{n}$, and is further normalised by the softmax function to get a correlation matrix $\beta$:

$$\beta_{i,j} = \frac{\exp(m_{i,j})}{\sum_{l=0}^{(H*W)-1} \exp(m_{i,l})}, \tag{2}$$

where $\beta_{i,j}$ represents the correlation value between the $i^{th}$ word and the $j^{th}$ subregion of the image. Then, the image subregion-aware word features $b \in \mathbb{R}^{D \times L}$ can be obtained by $b = \tilde{n}\beta^T$, which aggregates all spatial information weighted by the word-context correlation matrix $\beta$.

Additionally, to further reduce the negative impact of less important words, we adopt the word-level self-attention [12] to derive a 1D vector $\gamma$ with length $L$ reflecting the relative importance of each word. Then, we repeat $\gamma$ by $D$ times to produce $\gamma'$, which has the same size as $b$. Next, $b$ is further reweighted by $\gamma'$ to get $\tilde{b}$, denoted as $\tilde{b} = b \odot \gamma'$, where $\odot$ represents element-wise multiplication. Finally, we derive the correlation between the $i^{th}$ word and the whole image as Eq. (3):

$$r_i = \sigma\left(\frac{(\tilde{b}_i)^T w_i}{||\tilde{b}_i|| \, ||w_i||}\right), \tag{3}$$

where $\sigma$ is the sigmoid function, $r_i$ evaluates the correlation between the $i^{th}$ word and the image, and $\tilde{b}_i$ and $w_i$ represent the $i^{th}$ column of $b$ and $w$, respectively.

Therefore, the final correlation value $\mathcal{L}_{\text{corre}}$ between image $I$ and sentence $S$ is calculated by summing all word-context correlations, denoted as $\mathcal{L}_{\text{corre}}(I, S) = \sum_{i=0}^{L-1} r_i$. By doing so, the generator can receive fine-grained feedback from the word-level discriminator for each visual attribute, which can further help supervise the generation and manipulation of each subregion independently.

### 3.4 Perceptual Loss

Without adding any constraint on text-irrelevant regions (e.g., backgrounds), the generated results can be highly random, and may also fail to be semantically consistent with other content. To mitigate this randomness, we adopt the perceptual loss [7] based on a 16-layer VGG network [21] pre-trained on the ImageNet dataset [18]. The network is used to extract semantic features from both the generated image $I'$ and the real image $I$, and the perceptual loss is defined as

$$\mathcal{L}_{\text{per}}(I', I) = \frac{1}{C_i H_i W_i} \|\phi_i(I') - \phi_i(I)\|_2^2, \tag{4}$$

where $\phi_i(I)$ is the activation of the $i^{th}$ layer of the VGG network, and $H_i$ and $W_i$ are the height and width of the feature map, respectively.

To our knowledge, we are the first to apply the perceptual loss [7] in controllable text-to-image generation, which can reduce the randomness involved in the image generation by matching feature space.

### 3.5 Objective Functions

The generator and discriminator are trained alternatively by minimising both the generator loss $\mathcal{L}_G$ and discriminator loss $\mathcal{L}_D$.

**Generator objective.** The generator loss $\mathcal{L}_G$ as Eq. (5) contains an adversarial loss $\mathcal{L}_{G_k}$, a text-image correlation loss $\mathcal{L}_{\text{corre}}$, a perceptual loss $\mathcal{L}_{\text{per}}$, and a text-image matching loss $\mathcal{L}_{\text{DAMSM}}$ [25].

$$\mathcal{L}_G = \sum_{k=1}^{K} (\mathcal{L}_{G_k} + \lambda_2 \mathcal{L}_{\text{per}}(I_k', I_k) + \lambda_3 \log(1 - \mathcal{L}_{\text{corre}}(I_k', S))) + \lambda_4 \mathcal{L}_{\text{DAMSM}}, \tag{5}$$

where $K$ is the number of stages, $I_k$ is the real image sampled from the true image distribution $P_{\text{data}}$ at stage $k$, $I_k'$ is the generated image at the $k^{th}$ stage sampled from the model distribution $PG_k$,

$\lambda_2, \lambda_3, \lambda_4$ are hyper-parameters controlling different losses, $\mathcal{L}_{\text{per}}$ is the perceptual loss described in Sec. 3.4, which puts constraint on the generation process to reduce the randomness, the $\mathcal{L}_{\text{DAMSM}}$ [25] is used to measure text-image matching score based on the cosine similarity, and $\mathcal{L}_{\text{corre}}$ reflects the correlation between the generated image and the given text description considering spatial information.

The adversarial loss $\mathcal{L}_{G_k}$ is composed of the unconditional and conditional adversarial losses shown in Eq. (6): the unconditional adversarial loss is applied to make the synthetic image be real, and the conditional adversarial loss is utilised to make the generated image match the given text $S$.

$$\mathcal{L}_{G_k} = \underbrace{-\frac{1}{2}E_{I_{k'}\sim PG_k}\left[\log(D_k(I_k'))\right]}_{\text{unconditional adversarial loss}} \underbrace{-\frac{1}{2}E_{I_{k'}\sim PG_k}\left[\log(D_k(I_k', S))\right]}_{\text{conditional adversarial loss}}. \tag{6}$$

**Discriminator objective.** The final loss function for training the discriminator $D$ is defined as:

$$\mathcal{L}_D = \sum_{k=1}^{K}(\mathcal{L}_{D_k} + \lambda_1(\log(1 - \mathcal{L}_{\text{corre}}(I_k, S)) + \log\mathcal{L}_{\text{corre}}(I_k, S'))), \tag{7}$$

where $\mathcal{L}_{\text{corre}}$ is the correlation loss determining whether word-related visual attributes exist in the image (see Sec. 3.3), $S'$ is a mismatched text description that is randomly sampled from the dataset and is irrelevant to $I_k$, and $\lambda_1$ is a hyper-parameter controlling the importance of additional losses.

The adversarial loss $\mathcal{L}_{D_k}$ contains two components: the unconditional adversarial loss determines whether the image is real, and the conditional adversarial loss determines whether the given image matches the text description $S$:

$$\begin{aligned}\mathcal{L}_{D_k} = &\underbrace{-\frac{1}{2}E_{I_k\sim P_{\text{data}}}\left[\log(D_k(I_k))\right] - \frac{1}{2}E_{I_{k'}\sim PG_k}\left[\log(1 - D_k(I_k'))\right]}_{\text{unconditional adversarial loss}} \\ &\underbrace{-\frac{1}{2}E_{I_k\sim P_{\text{data}}}\left[\log(D_k(I_k, S))\right] - \frac{1}{2}E_{I_{k'}\sim PG_k}\left[\log(1 - D_k(I_k', S))\right]}_{\text{conditional adversarial loss}}.\end{aligned} \tag{8}$$

# 4 Experiments

To evaluate the effectiveness of our approach, we conduct extensive experiments on the CUB bird [23] and the MS COCO [10] datasets. We compare with two state of the art GAN methods on text-to-image generation, StackGAN++ [28] and AttnGAN [25]. Results for the state of the art are reproduced based on the code released by the authors.

## 4.1 Datasets

Our method is evaluated on the CUB bird [23] and the MS COCO [10] datasets. The CUB dataset contains 8,855 training images and 2,933 test images, and each image has 10 corresponding text descriptions. As for the COCO dataset, it contains 82,783 training images and 40,504 validation images, and each image has 5 corresponding text descriptions. We preprocess these two datasets based on the methods introduced in [27].

## 4.2 Implementation

There are three stages ($K = 3$) in our ControlGAN generator following [25]. The three scales are $64 \times 64$, $128 \times 128$, and $256 \times 256$, and spatial and channel-wise attentions are applied at the stages 2 and 3. The text encoder is a pre-trained bidirectional LSTM [20] to encode the given text description into a sentence feature with dimension 256 and word features with length 18 and dimension 256. In the perceptual loss, we compute the content loss at layer relu2_2 of VGG-16 [21] pre-trained on the ImageNet [18]. The whole network is trained using the Adam optimiser [8] with the learning rate 0.0002. The hyper-parameters $\lambda_1$, $\lambda_2$, $\lambda_3$, and $\lambda_4$ are set to 0.5, 1, 1, and 5 for both datasets, respectively.

Table 1: Quantitative comparison: Inception Score, R-precision, and $L_2$ reconstruction error of state of the art and ControlGAN on the CUB and COCO datasets.

| Method | CUB | | | COCO | | |
|---|---|---|---|---|---|---|
| | IS | Top-1 Acc(%) | $L_2$ error | IS | Top-1 Acc(%) | $L_2$ error |
| StackGAN++ | $4.04 \pm .05$ | $45.28 \pm 3.72$ | 0.29 | $8.30 \pm .10$ | $72.83 \pm 3.17$ | 0.32 |
| AttnGAN | $4.36 \pm .03$ | $67.82 \pm 4.43$ | 0.26 | $\mathbf{25.89 \pm .47}$ | $\mathbf{85.47 \pm 3.69}$ | 0.40 |
| Ours | $\mathbf{4.58 \pm .09}$ | $\mathbf{69.33 \pm 3.23}$ | **0.18** | $24.06 \pm .60$ | $82.43 \pm 2.43$ | **0.17** |

## 4.3 Comparison with State of the Art

**Quantitative results.** We adopt the Inception Score [19] to evaluate the quality and diversity of the generated images. However, as the Inception Score cannot reflect the relevance between an image and a text description, we utilise R-precision [25] to measure the correlation between a generated image and its corresponding text. We compare the top-1 text-to-image retrieval accuracy (Top-1 Acc) on the CUB and COCO datasets following [12].

Quantitative results are shown in Table 1, our method achieves better IS and R-precision values on the CUB dataset compared with the state of the art, and has a competitive performance on the COCO dataset. This indicates that our method can generate higher-quality images with better diversity, which semantically align with the text descriptions.

To further evaluate whether the model can generate controllable results, we compute the $L_2$ reconstruction error [12] between the image generated from the original text and the one from the modified text shown in Table 1. Compared with other methods, ControlGAN achieves a significantly lower reconstruction error, which demonstrates that our method can better preserve content in the image generated from the original text.

**Qualitative results.** We show qualitative comparisons in Fig. 4. As we can see, according to modifying given text descriptions, our approach can successfully manipulate specific visual attributes accurately. Also, our method can even handle out-of-distribution queries, e.g., *red zebra on a river* shown in the last two columns of Fig. 4. All the above indicates that our approach can manipulate different visual attributes independently, which demonstrates the effectiveness of our approach in disentangling visual attributes for text-to-image generation.

Fig. 5 shows the visual comparison between ControlGAN, AttnGAN [25], and StackGAN++ [28]. It can be observed that when the text is modified, the two compared approaches are more likely to generate new content, or change some visual attributes that are not relevant to the modified text. For instance, as shown in the first two columns, when we modify the colour attributes, StackGAN++ changes the pose of the bird, and AttnGAN generates new background. In contrast, our approach is able to accurately manipulate parts of the image generation corresponding to the modified text, while preserving the visual attributes related to unchanged text.

In the COCO dataset, our model again achieves much better results compared with others shown in Fig. 5. For example, as shown in the last four columns, the compared approaches cannot preserve the shape of objects and even fail to generate reasonable images. Generally speaking, the results on COCO are not as good as on the CUB dataset. We attribute this to the few text-image pairs and more abstract captions in the dataset. Although there are a lot of categories in COCO, each category only has a few number of examples, and captions focus mainly on the category of objects rather than detailed descriptions, which makes text-to-image generation more challenging.

## 4.4 Component Analysis

**Effectiveness of channel-wise attention.** Our model implements channel-wise attention in the generator, together the spatial attention, to generate realistic images. To better understand the effectiveness of attention mechanisms, we visualise the intermediate results and corresponding attention maps at different stages.

We experimentally find that the channel-wise attention correlates closely with semantic parts of objects, while the spatial attention focuses mainly on colour descriptions. Fig. 6 shows several

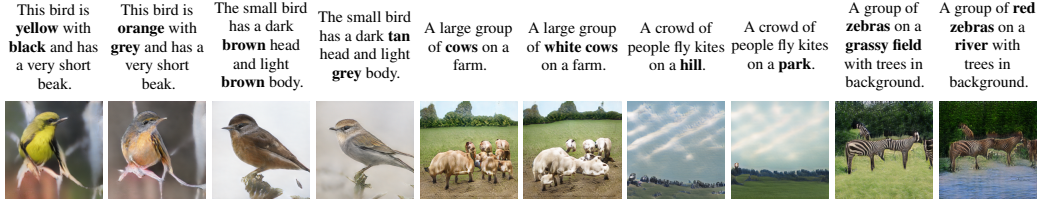

Figure 4: Qualitative results on the CUB and COCO datasets. Odd-numbered columns show the original text and even-numbered ones the modified text. The last two are an out-of-distribution case.

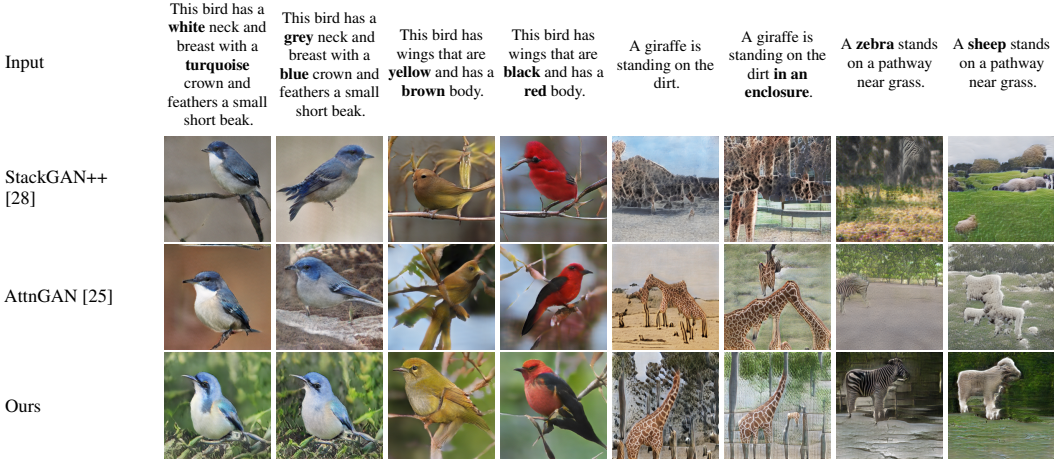

Figure 5: Qualitative comparison of three methods on the CUB and COCO datasets. Odd-numbered columns show the original text and even-numbered ones the modified text.

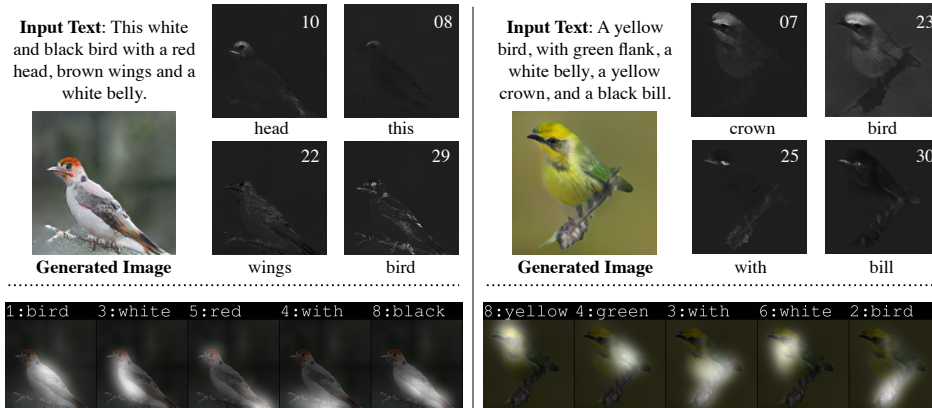

Figure 6: Top: visualisation of feature channels at stage 3. The number at the top-right corner is the channel number, and the word that has the highest correlation $\alpha_{i,j}$ in Eq. 1 with the channel is shown under the image. Bottom: spatial attention produced in stage 3.

channels of feature maps that correlate with different semantics, and our channel-wise attention module assigns large correlation values to channels that are semantically related to the word describing parts of a bird. This phenomenon is further verified by the ablation study shown in Fig. 7 (left side). Without channel-wise attention, our model fails to generate controllable results when we modify the text related to parts of a bird. In contrast, our model with channel-wise attention can generate better controllable results.

**Effectiveness of word-level discriminator.** To verify the effectiveness of the word-level discriminator, we first conduct an ablation study: our model is trained without word-level discriminator, shown in Fig. 7 (right side), and then we construct a baseline model by replacing our discriminator with

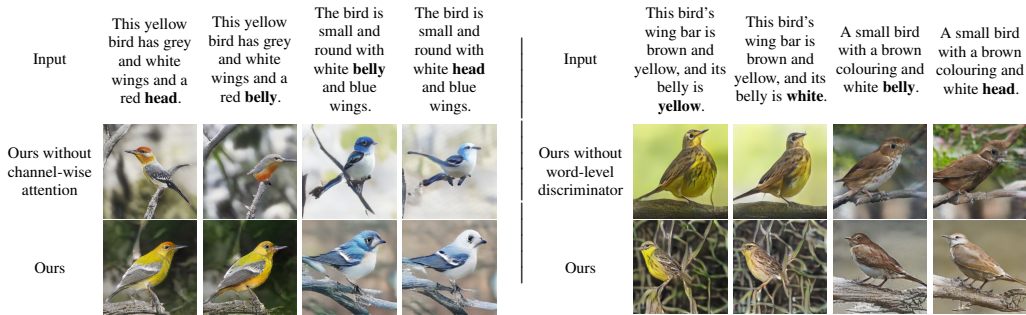

Figure 7: Left: ablation study of channel-wise attention; right: ablation study of the word-level discriminator.

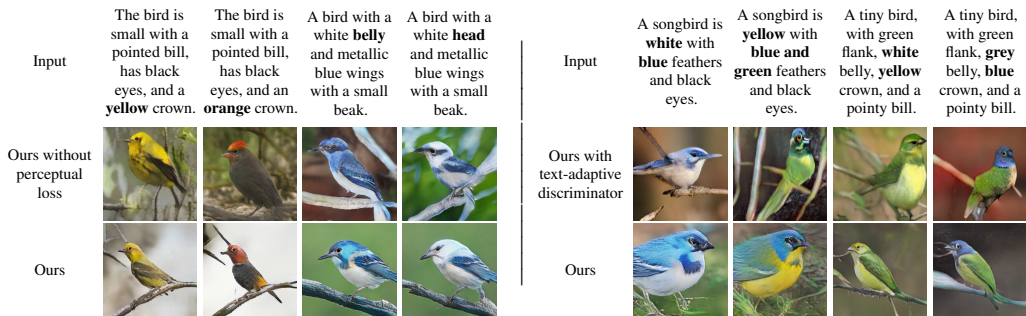

Figure 8: Left: ablation study of the perceptual loss [7]; right: comparison between our word-level discriminator and text-adaptive discriminator [12].

a text-adaptive discriminator [12], which also explores the correlation between image features and words. Visual comparisons are shown in Fig. 8 (right side). We can easily observe that the compared baseline fails to manipulate the synthetic images. For example, as shown in the first two columns, the bird generated from the modified text has a totally different shape, and the background has been changed as well. This is due to the fact that the text-adaptive discriminator [12] uses a global pooling layer to extract image features, which may lose important spatial information.

**Effectiveness of perceptual loss.** Furthermore, we conduct an ablation study: our model is trained without the perceptual loss, shown in Fig. 8 (left side). Without perceptual loss, images generated from modified text are hard to preserve content that are related to unmodified text, which indicates that the perceptual loss can potentially introduce a stricter semantic constraint on the image generation and help reduce the involved randomness.

## 5 Conclusion

We have proposed a controllable generative adversarial network (ControlGAN), which can generate and manipulate the generation of images based on natural language descriptions. Our ControlGAN can successfully disentangle different visual attributes and allow parts of the synthetic image to be manipulated accurately, while preserving the generation of other content. Three novel components are introduced in our model: 1) the word-level spatial and channel-wise attention-driven generator can effectively disentangle different visual attributes, 2) the word-level discriminator provides the generator with fine-grained training signals related to each visual attribute, and 3) the adoption of perceptual loss reduces the randomness involved in the generation, and enforces the generator to reconstruct content related to unmodified text. Extensive experimental results demonstrate the effectiveness and superiority of our method on two benchmark datasets.

**Acknowledgements.** This work was supported by the Alan Turing Institute under the UK EPSRC grant EP/N510129/1, the AXA Research Fund, the ERC grant ERC-2012-AdG 321162-HELIOS, EPSRC grant Seebibyte EP/M013774/1 and EPSRC/MURI grant EP/N019474/1. We would also like to acknowledge the Royal Academy of Engineering and FiveAI.

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
