[Supplementary Material]

# Supplementary Material

## 1 The Basic Architecture of the Generator

As shown in Figure 2, the network has a multi-stage cascaded architecture that contains $m$ generators $(G_0, G_1, ..., G_{m-1})$. Each stage takes a hidden state $b_i$ generated from the previous stage as input, and produces images $v_i$ of small to large scales. The first hidden state is the sentence embedding $s$ that is encoded by a pre-trained bidirectional RNN [7]. We also use the conditioning augmentation method (CA) [6] to smooth over text representation and to encourage robustness to small perturbation along the conditioning manifold:

$$
\begin{aligned}
b_0 &= F_0(z, F_{CA}(s)), \\
b_i &= F_i(b_{i-1}, F_{attn_i}(b_{i-1}, w, F_{CA}(s))), i = 1, 2, ..., m-1, \\
v_i &= G_i(b_i),
\end{aligned}
\tag{1}
$$

where $z \sim N(0, 1)$ denotes random noises, $w$ is the word embeddings, $F_{attn_i}$ is proposed word-level spatial and channel-wise attention model including two components: spatial attention model $S_{Attn_{i-1}}$ and channel-wise attention model $C_{Attn_{i-1}}$, and $F_0$, $F_i$, $G_i$ are denoted as neural networks. Then,

$$
F_{attn_i}(b_{i-1}, w, F_{CA}(s)) = concat(S_{Attn_{i-1}}, C_{Attn_{i-1}}).
\tag{2}
$$

## 2 Spatial Attention Model

The spatial attention model takes two inputs: the word embeddings $w$ and the visual features $v$ from the previous stage. Then, by using a perception layer $F$, the word embeddings $w$ are converted into the common semantic space of the visual features denoted as $\tilde{w} = Fw$.

Next, we compute the dot product between $\tilde{w}$ and visual features $v$ to get a word-context vector, which is followed by the *Softmax* function to produce the attention weights. Thus, the spatial attentive word-context feature $c$ is obtained by computing the dot-product between the attention weights and $\tilde{w}$:

$$
c_i = \sum_{j=0}^{T-1} \alpha_{i,j} \tilde{w}_j, \quad \text{where } \alpha_{i,j} = \frac{e^{s_{i,j}}}{\sum_{k=0}^{T-1} e^{s_{i,k}}}, s = \tilde{w}^T v.
\tag{3}
$$

Here, $\alpha_{i,j}$ represents the correlation between the $i^{th}$ sub-region of the image and the $j^{th}$ word. Thus, the $c_i$ indicates the correlation between the $i^{th}$ sub-region of the image and the whole sentence.