[Reviews · NeurIPS 2019]

Reviewer 1



Positive Aspects 1. The paper is well-organized and written, which can be followed easily. 2. The considered problem is interesting. In particular, instead of generating a new image from the text, the authors pay more attention to image manipulation based on the modified natural language description. 3. The visual results seem very impressive, which are able to manipulate the parts of images accurately according to the changed description while preserving the unchanged parts. Negative Aspects 1. For the word-level spatial and channel-wise attention driven generator: (1) The novelty and effectiveness of attentional generator may be limited. Specifically, the paper designs a word-level spatial and channel-wise attention driven generator, which has two attention parts (i.e. channel-wise attention and spatial attention). However, since the spatial attention is based on the method in AttnGAN [7], most contributions may lie on the additional channel-wise part. But instead of verifying the impact of channel-wise attention, the paper reports the visual attention maps (Fig.5) according to the integration of channel-wise and spatial attention. Moreover, the visualized maps in Fig.5 are similar to the visual results in AttnGAN [7] and thus cannot prove the effectiveness of the proposed channel-wise attention method. (2) In Fig.6, it seems that the generation image without attention mechanism (SPM only) can also achieve desired visual performance (i.e., the generated image only changes the colour of the bird while keeping the rest). Thus, I am not sure whether the attention mechanism is really necessary for the proposed method. 2. For word-level discriminator: (1) In line 132-133, it mentions that “Additionally, to further reduce the negative impact of less important words, we apply a word-level self-attention mechanism”. More experiments should be conducted to verify the influence of the additional self-attention. Otherwise, it cannot demonstrate whether the self-attention mechanism is necessary for the discriminator and how it impacts the final results. (2) Moreover, to prove the effectiveness of the proposed word-level discriminator, it would be better to conduct an ablation study with and without this discriminator. 3. For semantic preservation model (SPM): The proposed SPM is an application of the perceptual loss [24], where the novelty is also slight. 4. Minor issues: Line 231, “Fig.4 also shows the …” should be “Fig.5 also shows the …”. Final remarks: For this paper, I have mixed feelings. In positive, the studied problem is interesting and the generated results are impressive. Nevertheless, on the other hand, since there are not sufficient experiments to verify the impact and necessity of each proposed component, it is not clear whether they are significant or not in this task. Besides, the novelty of some proposed parts may be limited (e.g. SPM).

Reviewer 2



- How does the channel wise attention module help in addition to the word-spatial attention? An ablation here would be useful to validate this architecture choice. - In equation 4, what is the interpretation of \gamma_i. Since the word embeddings come from a pre-trained bidirectional RNN, it is not clear why this number should correspond to a notion of the “importance” of a word in the sentence, and it is not clear to me what importance means here. Is my understanding correct that the objective in equation 5 is for the model to maximize the correlation between words depicted in the image and the important words in the sentence? - When the semantics preservation model is used, how does this affect the diversity of the model samples? To my knowledge the proposed word-level discriminator and channel-wise attention are novel. The paper is well written despite a few unclear points mentioned above. However, the significance is difficult to judge because the CUB dataset is somewhat saturated, MS-COCO samples are not realistic enough to easily assess the controllability, and the text queries do not seem out of sample enough to test the limits of the text controllability.

Reviewer 3



Paper summary: This paper proposed a novel text-to-image synthesis method with its focus on learning a representation sensitive to different visual attributes and better spatial control from the text. To achieve this goal, a novel generator with word-level spatial and channel-wise attention model has been adopted based on the correlation between words and feature channels. To further encourage the word-level control, a novel correlation loss defined between words and image regions has been explicitly enforced during model learning. Experimental evaluations have been conducted on benchmark datasets including CUB and COCO. Overall, this paper is an interesting extension to the text-to-image synthesis and multi-modal representation learning. In general, it is clearly written with reasonable formulation and cool qualitative results. However, reviewer does have a few concerns regarding the current draft. - Current title is ambitious or not very precise as text-to-image generation is often coupled with a learned controllable representation. - What is the shape of visual features v? As far as reviewer can see, in Equation (1), v indicates the feature map with single image channel; in Equation (2), v indicates the feature map within image sub-region. It would be great if this can be clarified in the rebuttal. - Equation (6): j is defined but not used. - Table 1 and Figure 3: While reviewer is impressed by the results on CUB, results on COCO don’t look good. Also, it seems like the model only learns to be sensitive to the color but not shape or other factors. - Figure 4: On CUB, two images generated by similar text (with one concept difference) look almost identitcal (e.g., same shape, similar background color). On COCO, It is clear that two images generated by similar text (with one concept difference) look a bit different (see last four columns). Reviewer would like to know which loss function enforces such property. Why the same loss completely fails on COCO?

[Author Response · NeurIPS 2019]



Figure 1: Left: visualisation of feature channels. The number on the right top corner is the channel number. The word that has the highest correlation $\alpha_{i,j}$ in Eqn.1 with the channel is shown under the image. Right: ablation study.

Figure 2: Left: ablation study. Right: Qualitative comparison of two methods on the COCO dataset. Odd-numbered columns show the original text and even-numbered ones the modified text. Please zoom in to see clearer.

**R1: The effectiveness of channel-wise attention.** We experimentally find that the channel-wise attention highly correlates with semantically meaningful parts while the spatial attention focuses on colour descriptions. Fig.1(L) shows several channels of feature maps transformed with our channel-wise attention. It shows that channels closely correlate to semantic parts, such as belly, eye, wing, crown. This phenomenon is further verified by the results shown in Fig.1(R). Without channel-wise attention, the model fails to generate controllable results when we modify the text related to parts of a bird. In contrast, our full model with channel-wise attention can generate much better controllable results.

**R1: The necessity of attention.** In the paper Fig. 6 and Tab. 2, *SPM only* means the model does not incorporate the channel-wise attention but still has the spatial attention. We will clarify this in our main paper. Fig. 6 *SPM only* verifies that spatial attention works well if only colour information is modified. Our channel-wise attention is responsible for parts related modification as described above.

**R1: Word-level discriminator.** The word-level discriminator can help better disentangle different visual attributes and facilitate image manipulation as shown in Fig.2(L). With word-level discriminator, our model achieves better results, e.g., the original shape and colour of the bird are well preserved.

**R1: Novelty of SPM.** To our best knowledge, we are the first to apply perceptual loss in controllable text-to-image generation and show its effectiveness on reducing randomness involved in generation, which makes manipulation stable.

**R2: Channel-wise attention.** Please refer to "R1: The effectiveness of channel-wise attention."

**R2: The importance of self-attention.** We adopt the pre-trained RNN used in AttnGAN. The objective function used to train RNN is to improve text-image matching score based on cosine similarity. Thus, it is reasonable to use self-attention to reduce impact from less important words which has small cosine similarity.

**R2: SPM and diversity of the model.** The SPM would not affect the diversity of the model, as the diversity comes from the random noise sampled from a normal distribution in stage 0. We will clarify this in our paper.

**R2: The controllability and out-of-distribution on COCO.** In this rebuttal, we focus on the animal subset of COCO. As shown in Fig.2, our model can effectively manipulate the images compared with AttnGAN and also work well on out of distribution queries, e.g., red zebras on the river.

**R3: Shape of the visual features, sensitivity.** $v$ in equation 1 indicates the generated image features whose shape is $B \times N \times (H * W)$, where $N$ is 32, $H * W$ is $64 * 64$ in stage 1 and $128 * 128$ in stage 2. The $v$ in equation 2 indicates real image features whose shape is same as generated images. We will clarify this.

**R3: CUB and COCO experiments** COCO has much more diversified descriptions where each only contains few examples. The low density of corresponding images is a reason for the different behaviours on the two datasets. Also, captions in COCO are more abstract and focus on the category of objects, which makes text-to-image generation be more challenging on COCO. However, we still produce much better results compared with AttnGAN shown in Fig.2(R).

[Meta-Review · NeurIPS 2019]

This paper was reviewed by three expert reviewers and received three Weak Accept recommendations. After rebuttal, all the reviewers are positive about this paper, and agree that the paper is generally well written, the considered problem is interesting, and the results are impressive. Nevertheless, on the other hand, both R1 and R2 commented that it is difficult to judge the significance of the results due to lack of sufficient ablation studies and the fact that CUB is saturated. Besides, R1 had concerns regarding the novelty of the paper, and R3 left several detailed comments for the authors to deal with. The rebuttal partially solves the reviewers' concerns. On balance, the AC recommends accepting the paper, but also strongly advise the authors to include additional comparisons and other revisions suggested by the reviewers and promised in the rebuttal.